# Multi-Gene Mutation Profiling by Targeted Next-Generation Sequencing in Premenopausal Breast Cancer

**DOI:** 10.3390/genes13081362

**Published:** 2022-07-29

**Authors:** Eleni Zografos, Angeliki Andrikopoulou, Alkistis Maria Papatheodoridi, Maria Kaparelou, Garyfalia Bletsa, Michalis Liontos, Meletios-Athanasios Dimopoulos, Flora Zagouri

**Affiliations:** 1Department of Clinical Therapeutics, Alexandra Hospital, School of Medicine, National and Kapodistrian University of Athens, 80 Vasilissis Sofias Avenue, 11528 Athens, Greece; el_zogra@hotmail.com (E.Z.); aggandrikop@med.uoa (A.A.); alkistispapath@gmail.com (A.M.P.); kapareloum@gmail.com (M.K.); mliontos@gmail.com (M.L.); mdimop@med.uoa.gr (M.-A.D.); 2Hellenic Anticancer Institute, 10680 Athens, Greece; bletsag@yahoo.gr

**Keywords:** premenopausal breast cancer, mutation, multigene panel testing, NGS

## Abstract

Breast cancer has distinct etiology, prognoses, and clinical outcomes at premenopausal ages. Determination of the frequency of germline and somatic mutations will refine our understanding of the genetic contribution to premenopausal breast cancer susceptibility. We applied a comprehensive next generation sequencing-based approach to analyze blood and/or tissue samples of 54 premenopausal breast cancer patients treated in our clinic. Genetic testing results were descriptively analyzed in correlation with clinicopathological data. In the present study, 42.5% of premenopausal breast cancer patients tested carried pathogenic mutations in cancer predisposition genes (*CHEK2*, *BRCA1*, *TP53*, and *MUTYH*). Germline variants of unknown/uncertain significance (VUSs) in eight different cancer susceptibility genes, namely *BRCA1*, *BRCA2*, *CHEK2*, *RAD51C*, *RAD51D*, *ATM*, *BRIP1*, and *PMS2*, were also identified in 14 premenopausal patients (35%). Of the breast tumors tested, 61.8% harbored pathogenic somatic variants in tumor suppressor genes (*TP53*, *NF1*, *RB*), genes involved in DNA repair (*BRCA1*, *BRCA2*, *ATM*, *RAD50*), cell proliferation (*PTEN*, *PIK3C FGFR3*, *AKT1*, *ROS1*, *ERBB2*, *NOTCH1*), and cell adhesion (*CTNNB1*). This descriptive study employs the powerful NGS technology to highlight the high frequency of premenopausal cases attributable to genetic predisposition. Mutation identification in a larger cohort may further ensure that these patients receive tailored treatment according to their menopausal status.

## 1. Introduction

Globally, female breast cancer has become the most commonly diagnosed cancer, with 2,261,419 new diagnoses and 684,996 deaths estimated to have occurred in 2020 [1]. Breast malignancy is considered a biologically and clinically heterogeneous entity, with several recognized risk factors, some of which are modifiable, while others non modifiable [2]. Among the latter ones, the inextricable link between ageing and breast cancer remains undisputed since only about 4–6% of cases are diagnosed in women aged younger than 40 years [3,4]. Unfortunately, premenopausal women, especially younger ones, face inferior survival, in part because they present with more aggressive clinicopathologic features and distinct age–incidence molecular profiles, including greater prevalence of endocrine-unresponsive tumors, higher histological grade at diagnosis, nodal involvement, and higher rates of Her2/neu overexpression [5,6,7,8]. Additionally, early identification of breast cancer is more challenging in premenopausal women due to increased breast density, with tumors often being detected at later stages [9]. Naturally, this age group is of particular interest from a public and patient health perspective, as they represent a socially active subset of the general population, and thus research efforts are geared towards ensuring the optimal individualized management for those patients.

Regarding the familial predisposition of the disease, genetic risk factors should be considered more in premenopausal patients, since a higher incidence of pathogenic germline mutations has been reported in younger women. In this setting, according to the latest international guidelines, hereditary cancer risk evaluation and testing is clinically indicated for all female breast cancer patients diagnosed under the age of 45 [10]. These women are, in their overwhelming majority, premenopausal, as epidemiological data for the timing of menopause in developed nations estimate a median age ranging from 48–52 years [11,12]. The hereditary component of premenopausal breast cancer is further illustrated by the considerably higher *BRCA1/2* mutation prevalence in families with a history of breast cancer and premenopausal female members affected by the disease [13]. Prior to 2020, the National Comprehensive Cancer Network (NCCN) Guidelines for Breast and Ovarian Cancer Genetic/Familial High-Risk Assessment mainly focused on testing criteria for *BRCA1/2* and determining appropriate risk management for carriers of pathogenic variants in these two genes [14]. However, cancer predisposing mutations are not limited to *BRCA*, with the frequency of pathogenic germline variants in known risk genes in young women being as high as 23% [15,16]. This newly acquired evidence that strongly highlights the clinical validity of protein truncating variants in genes beyond *BRCA1/2*, led to the expansion of these guidelines to include testing of other high-risk breast cancer genes, including *CHEK2*, *CDH1*, *BRIP1*, *NF1*, *PALB2*, *PTEN*, *RAD51C*, *RAD51D*, *STK11*, and *TP53* [17,18]. The inclusion of these genes in the genetic susceptibility evaluation is based on the results of either prospective cohort studies in a population-based setting or of multiple traditional retrospective case-control studies that have demonstrated increased breast cancer risk and whose findings are collectively supported by a uniform consensus [10].

Nowadays, clinical testing with panels of multiple genes has become easily accessible, especially with the widespread use of affordable sequencing. In this context, most young breast cancer patients are offered the option of multigene panel testing, which does not only include high-penetrance susceptibility genes but can also encompass genes associated with lower penetrance. Genetic risk evaluation can have a substantial impact on ensuring the individualized management of this cohort since it can help distinguish DNA damage response deficient subgroups of patients who may benefit from platinum-based chemotherapy [19] and poly ADP-ribose polymerase [PARP] inhibition [20]. Additionally, germline genetic testing offers the potential for establishing targeted follow-up protocols that will ensure early detection of subsequent tumors that are more frequent in high-risk gene carriers [21]. Early genetic diagnosis offers a significant benefit not only for the affected patient, but also for at-risk family members to enable genetically targeted disease prevention (chemoprevention and/or prophylactic surgeries) [22]. It further allows carriers of pathogenic variants to make decisions regarding family planning, which is an aspect of great importance for the psychosocial well-being of this age group [23].

Despite the paramount implication of genetic testing and the knowledge of inherited cancer susceptibility in breast cancer patients with younger age of onset, the genetic background of the disease in premenopausal women remains an understudied field. In the present descriptive study, we applied multiple-gene panel testing to cancer susceptibility genes using next generation sequencing (NGS)-based technology aiming to identify the proportion of premenopausal breast cancer cases attributable to genetic predisposition and to assess the prevalence of germline (i.e., inherited) and somatic mutations in women treated in our clinic prior to their menopause.

## 2. Materials and Methods

### 2.1. Patients

In this retrospective cohort study, incident cases of 54 premenopausal breast cancer patients that underwent Next Generation Sequencing (NGS) in blood samples and/or paraffin blocks were collected. All women attended the Oncology Department of the National and Kapodistrian University of Athens, at the “Alexandra” Hospital in Athens, Greece between January 2019 and December 2021. The sample pool derived from the hospital’s patient database, from which consecutive cases satisfying the abovementioned inclusion criteria were selected. The study was performed in accordance with the Helsinki Declaration and was approved by the Institutional Review Board (IRB) of the participating hospital. Participation was voluntary and, once informed consent was granted by each of the eligible patients, medical files of the participants were reviewed; researchers collected demographic and clinical data, including age at diagnosis, family cancer history, genetic testing results, and histopathologic evaluation (tumor stage, size, grade, lymph node status, hormone receptor, ki67 expression, and HER2 status). The recommended definition of ER and PR positivity on the AJCC Staging Manual (8th edition) is 1% or more of cells that score positive by immunostaining [24]. Finally, a blood sample was collected from each premenopausal breast cancer patient, along with archival formalin fixed paraffin embedded (FFPE) breast tissue deriving from either mastectomy or breast conserving surgery prior to adjuvant treatment, whenever FFPE samples were available.

### 2.2. DNA Extraction and Targeted Next Generation Sequencing

Concerning germline targeted sequencing, genomic deoxyribonucleic acid (DNA) was isolated from whole blood samples that were collected in EDTA-containing tubes. Following centrifugation for 10 min at 3000 rpm at room temperature, each plasma sample was collected, given a unique identifier, and stored at −80 °C until use. Isolation of plasma DNA was performed using the QIAsymphony DSP DNA Mini Kit (Qiagen, Germantown, MD, USA) and indexed genomic libraries were prepared to target the sequence of 94 cancer predisposing genes using the Trusight™ Comprehensive Hereditary Cancer Panel—Nextera™ DNA Flex Pre-enrichment Library Prep (Illumina, San Diego, CA, USA). Libraries were qualitatively and quantitatively evaluated using a Fragment Analyzer (Advanced Analytical Technologies, Heidelberg, Germany) and sequenced on a MiSeq, NextSeq 500, or Hiseq 2500 sequencing platform (Illumina Inc, San Diego, CA, USA), where coding regions of the tested genes had a minimum base and amplicon coverage of 30× for the 96.2% of the targeted regions.

For the somatic analysis research, 35 paraffin-embedded breast tissues were cut at slices of 10 μm thickness. Tumor DNA was isolated from tissue samples using the QIAamp DNA FFPE Tissue (Qiagen, Germantown, MD, USA) or the Ion Ampliseq Custom Next Generation Sequencing (NGS) DNA panel (Thermo Fisher Scientific, Waltham, MA, USA), following the manufacturer’s protocol. Tissues then underwent DNA extraction via the QIAamp DNA FFPE Tissue, and libraries were constructed employing the AmpliSeq for Illumina Comprehensive Panel v3, that offers coverage of key cancer-associated genes. The NGS study was performed using an Illumina platform (MiSeq, NextSeq500 or NovaSeq) with a median amplicon coverage of 300× for the 93.4% of the targeted regions. In cases where DNA extraction was executed via the Ion Ampliseq kit, targeted NGS was performed using the Ion GeneStudio S5 Prime semiconductor-based system (Thermo Fisher Scientific, Waltham, MA, USA) with a median amplicon cover of 2000×. Subsequently, the Ion Reporter™ Software (Thermo Fisher Scientific, Waltham, MA, USA) was used for streamlined data analysis and variant calling against the human reference genome GRCh38. Additionally, data were curated manually using the genomic analysis software tool Oncomine Reporter (Thermo Fisher Scientific, Waltham, MA, USA) and the available catalogs of genomic variation observed in the germline as well as those that appear in tumors as somatic mutations (dbSNP, COSMIC, ClinVar, gnomAD etc.). All sequenced variants were filtered and evaluated based on the recommendations published by the American College of Medical Genetics and Genomics (ACMG) [25] and the NCCN guidelines [26]. More specifically, identified alterations were characterized as pathogenic when classified as disease causing or as variants of unknown significance (VUS) when evidence regarding their pathogenicity was either conflicting or limited, according to the criteria recommended to describe variants identified in Mendelian disorders by ACMG [25].

### 2.3. Statistical Analysis

All analyses were descriptive, according to the methodological features of an observational study with a non-interventional design. Categorical variables were displayed as frequency tables (N, %). The statistical analysis was carried out using the SPSS software.

## 3. Results

### 3.1. Characteristics of the Study Cohort

Histopathological features of the 54 premenopausal breast tumors are presented in Table 1. The median age of the patients was 40 years (SD = 6.7). The predominant histological type among patients enrolled in our study was invasive ductal carcinoma (IDC) (96.3%). The majority of tumors was of high grade (68.5%), and pathological measurements regarding tumor size revealed that 48.2% of them were above 2 cm (T2, T3, T4). Concerning receptor status, 77.8% of premenopausal breast cancer cases were ER positive, 74.1% PR positive, and 72.2% of them HER2 negative. Of note, 13.0% of the participants’ tumors were classified as triple receptor-negative breast cancer (TNBC), with the Luminal B, HER2-negative subtype being the prevalent one in our Greek cohort (35.2%). Axillary node infiltration was identified in 61.1% of cases. Additionally, 38.9% of our premenopausal patients reported a positive family history of malignancy, defined as ≥ 1 first- or second-degree relatives having a recorded cancer diagnosis.

### 3.2. Germline and Somatic Mutations

Fifty-four premenopausal patients with breast carcinoma that underwent genetic testing via NGS panels for either germline mutations (40), somatic mutations (34), or both (20), were included in this study. For those with a positive mutation result from genetic testing, Table 2 shows the identified germline mutations in cancer predisposition genes and the exact location of each mutation in the affected gene, along with the somatic mutational pattern of the premenopausal patients whose tumor samples we were able to obtain.

The detection rate of germline mutations among the cases tested was 42.5% (17/40). Regarding germline pathogenic variants, *CHEK2* was mutated in three out of seventeen premenopausal women, followed by *BRCA1* (2 of 17), *TP53* (1 of 17), and *MUTYH* (1 of 17). Interestingly, none of our subjects were identified to have the same pathogenic germline mutation. However, several of our premenopausal breast cancer patients carried germline variants of unknown/uncertain significance (VUSs) in eight different breast cancer susceptibility genes, namely *BRCA1*, *BRCA2*, *CHEK2*, *RAD51C*, *RAD51D*, *ATM*, *BRIP1*, and *PMS2*. Additionally, five out of the twenty patients tested for both intrinsic and acquired genomic alterations were carriers of concurrent germline and somatic variants.

The complete list of somatic variants identified in our study cohort, both pathogenic and of unknown significance, is presented in Table 2. Pathogenic somatic mutations were identified via NGS in 21 out of the 34 premenopausal breast tumors tested (61.8%). Notably, *TP53* (10 of 21; 47.6%) and *PIK3CA* (8 of 21; 38.1%) were the genes that most frequently harbored pathogenic somatic mutations in premenopausal cases, followed by *PTEN* (4 of 21), *BRCA2* (1 of 21), *AKT1* (1 of 21), *ATM* (1 of 21), and *ERBB2* (1 of 21). Collectively, all germline and somatic mutations of premenopausal women with breast cancer as determined by NGS in relation to each patient’s distinct clinical characteristics and histological staging is presented in greater detail in Appendix A.

## 4. Discussion

We here retrospectively identified 54 premenopausal breast cancer patients treated in our clinic, unselected for family history, ethnicity, or subtype. Of these, 40 patients were tested via comprehensive targeted next generation sequencing and 42.5% (17/40) of them were identified as carriers of germline mutations in cancer susceptibility genes. Additionally, somatic mutation analysis in 34 premenopausal breast cancer FFPE samples deriving from our cohort, revealed that 71.4% (25/34) of tested patients were carriers of somatic variations in tumor suppressor genes (*TP53*, *NF1*, *RB*), genes involved in DNA repair (*BRCA1*, *BRCA2*, *ATM*, *RAD50*), in cell proliferation (*PTEN*, *PIK3CA, FGFR3*, *AKT1*, *ROS1*, *ERBB2*, *NOTCH1*), and in cell adhesion (*CTNNB1*). To date, little available data on the frequency and spectra of germline and somatic mutations is specific to premenopausal women with breast cancer, despite being important for ascertaining the genomic background of this younger subset of patients.

The testing of germline mutations alongside somatic alterations is rapidly evolving as an integral part of precision-medicine therapy of patients with cancer. In fact, research hypotheses state that a large fraction of cancer predisposition genes, defined as genes in which germline mutations confer highly or moderately increased risks of cancer, could be oncogenic when mutated somatically [27]. In our descriptive study, 25% (5/20) patients tested for both intrinsic and acquired genomic alterations were carriers of concurrent germline and somatic variants. This observation raises the inevitable question of whether there are any associations between inherited breast cancer susceptibility loci and somatic mutations acquired de novo by breast cancer cells, especially in younger premenopausal patients where the genetic element is more prominent. Recent evidence suggest links between common germline risk variants and somatic mutations in genes that confer selective breast tumor growth advantage [28]. In the effort to understand the interaction of germline and somatic mutations, one could not fail to mention the infamous “two-hit” cancer hypothesis first proposed by Knudson in 1971 [29]. According to this approach, the first hit could be an inherited susceptibility variant via the germinal cells or a nonhereditary somatic mutation in an important cancer predisposition gene; the second hit occurs in somatic cells. Consequently, it may take fewer stages for individuals with strong genetic predisposition to develop breast cancer, in comparison to persons with reduced hereditary risk. On the other hand, in some cases presence of germline mutations may not be related to tumorigenesis either due to zygosity-dependent phenotype penetrance or loss of the pathogenic germline allele somatically [30]. In our cohort, only one patient had a coexisting pathogenic germline mutation and a somatic second hit, interestingly in the same gene (*TP53*), but on different loci. The effect of the simultaneous presence of germline and somatic mutations needs to be further addressed in a larger cohort of patients.

Especially when it comes to hormone sensitive tumors, significant differences in gene expression and somatic mutation patterns potentially driven by altered hormone levels have been described between premenopausal and postmenopausal breast cancer, revealing a menopausal status-dependent role of certain genes [31]. In total, 47.6% of premenopausal breast cancer samples carried somatic mutations in *TP53* and 38.1% in PIK3CA, the two most-frequently mutated genes according to the literature [32]. This finding is in line with a targeted deep sequencing analysis on premenopausal breast cancer patients of Latin American origin, which also pinpointed *TP53* as the most frequently mutated gene, followed by *PIK3CA* [33]. The same study reports a clinical association between somatic *TP53* mutations and the HER2-enriched molecular subtype, as it has also been shown by others either in the premenopausal population [34] or even irrespective of menopausal status [35]. This correlation has been attributed to the fact that p53 mutants can potentially induce HER2 up-regulation and favor the stabilization of the protein [36]. Most of these somatic mutations are located in residues corresponding to the DNA-binding domain of *TP53* and result in decreased DNA-binding affinity and gene transactivation, thus playing a role in the early onset and prognosis of breast cancer [37]. Of the ten exonic mutations observed in this study, nine are in the DNA-binding domain of the TP53 gene, whereas seven out of ten tumors tested positive for HER2 expression, despite only two premenopausal cases falling under the HER2/*neu* subtype. Additionally, “acquired” mutations of the oncogene *PIK3CA* identified in tumors of our cohort after the implementation of a custom next generation sequencing DNA panel are commonly found in Luminal/ER-positive tumors, that comprise the majority of our cases (77.8%). The benefit of alpelisib in *PIK3CA*-mutated, hormone-receptor-positive, HER2-negative advanced breast cancer highlights the increasing clinical importance of *PIK3CA* testing [38]. Although the indication of PI3Kα-selective inhibitor and degrader is limited to postmenopausal patients, ongoing clinical trials enroll postmenopausal women that meet the abovementioned inclusion criteria [39]. So far, reports on the somatic genomic landscape of breast cancer state that *PIK3CA* mutations are less frequent in young women when compared with older ones [40]. Whether the level of PI3K pathway activity correlates etiologically with age and/or menopause status bears further investigation. The remaining cases displayed a variable array of low-frequency mutations in distinct combinations (*NF1*, *RB*, *BRCA2*, *ATM*, *RAD50*, *PTEN*, *FGFR3*, *AKT1*, *ROS1*, *ERBB2*, *NOTCH1*, *CTNNB1*), as previously reported [41]. This observation draws attention to the fact that oncogenic interactions between genes harboring mutations is not a “straight-line” process but is largely defined by complex arrays of closely intertwined molecular networks and pathways.

Regarding pathogenic germline variants in premenopausal women, they were all detected in genes involved in DNA damage repair signaling, with *CHEK2* being mutated in three of seventeen premenopausal patients, followed by *BRCA1* (2 of 17), *TP53* (1 of 17), and *MUTYH* (1 of 17). (3/17), followed by *BRCA1* (2/17), *TP53* (1/17), and *MUTYH* (1/17). In our premenopausal cohort, the frequency of *BRCA1* mutations (11.8%) is similar to the prevalence identified in young breast cancer subsets by other investigators [42]. Interestingly, it has been shown that premenopausal *BRCA*-mutated patients with breast cancer are more likely to have more aggressive disease and face recurrences than postmenopausal carriers, underlying the fact that menopausal status should be taken into consideration as a potential prognostic factor in *BRCA* affected patients [43]. Furthermore, the autosomal inherited *TP53* gene found on chromosome 17p13.1 is an established causative factor for Li-Fraumeni syndrome [44]. Carriers of this mutation have a high lifetime cumulative risk of developing multiple malignancies and a strong family history of early-onset cancer. This is consistent with the hereditary profile of our patient that carried the p.C275Y pathogenic mutation (also known as c. 824G>A), located in coding exon 7 of *TP53*. Specifically, she was diagnosed at the very young age of 32 with HER2-enriched breast cancer and had a first-degree family history, with a mother affected by uterine sarcoma and a father by hepatocellular carcinoma. However, other studies focusing on the premenopausal population underline that early-onset breast cancer cannot be attributed to *TP53* polymorphisms alone, although specific mutations occur in this young cohort in higher frequency, without, however, always affecting p53 transactivation function [45]. Concerning *CHEK2*, we identified by a targeted gene sequencing panel three premenopausal patients who carried germline pathogenic mutations that affect the expression of this key cell cycle checkpoint kinase. Previous studies note an approximately 4% incidence of *CHEK2* *1100delC truncation carriers among premenopausal breast cancer patients [46]. This high-risk allele is associated with greater than twofold increase in risk and poor prognosis, but it was not detected in our cohort. This discrepancy may be due to the fact that the *CHEK2* c. 1100delC allele is rarely identified in breast cancer patients of Greek descent [47]. Pathogenic variants in *CHEK2* are associated with an increased risk of estrogen receptor-positive breast cancer, although the risk decreases significantly with age, making these carriers more likely to develop the disease prior to menopause [48,49]. Notably, the I157T low-risk allele that was identified in two of our patients is associated with moderate risk and a more favorable prognosis [50]. Lastly, *MUTYH* mutations that are known to predispose to recessively inherited colorectal polyposis and cancer have also been associated with breast cancer susceptibility [51,52], although there are some contradictory findings published [53]. Our results are consistent with previous studies that reported these mutations in young women diagnosed with invasive breast cancer who were advised to undergo frequent colonoscopy [16]. All in all, identification of germline mutations and variants in larger patient cohorts is warranted, as an optimal approach to unravel the underlying heritability of premenopausal breast cancer.

To perform genetic variation screening in premenopausal women with breast cancer we employed next generation sequencing (NGS) technologies in the form of multigene panels. There is a current trend in incorporating these technologies in everyday medical care to guide treatment choices and adopt an integrated counselling approach for each patient. However, using these technologies in clinical practice on a regular basis presents a number of issues, one of which is determining how to interpret variants of unknown/uncertain significance (VUS) [54,55]. These variations in the genetic sequence, including single nucleotide polymorphisms or amino acid insertions/deletions, cause doubt for clinicians on how to properly advise patients since the association with cancer risk is unclear. Our results further illustrate that genetic testing often yields ambiguous results since 35% (14/40) of our premenopausal patients were carriers of germline VUSs, one in *BRCA1*, five in *BRCA2*, three in *CHEK2*, two in *RAD51*, one in *ATM,* one in *BRIP1*, and one in *PMS2*. These VUSs are alterations that may not even influence the function of the encoded protein; hence they should not be used to justify a change in clinical management, but should be managed based on the cancers present in the family [26]. Since the pathogenicity of germline variants is based on the currently available epidemiologic and functional data, a clinical takeaway of our germline sequencing findings is the significance of repeat germline testing as clinical techniques improve and panel sizes grow, especially for individuals at high germline risk, such as young premenopausal women.

Concerning the limitations of our analysis, one should not fail to mention the small sample size, which limits the power and generalizability of our work. Another downside regarding our methodology is the heterogeneity in the data sets in terms of age, molecular subtype, and other clinicopathological characteristics that made direct comparison somewhat challenging but not unfeasible, due to the exploratory nature of this descriptive study. Further collection of sufficiently large populations of premenopausal patients for the initial discovery and the subsequent validation is required to fully determine the prevalence of germline and somatic variants, particularly low frequency ones. Additionally, larger case–control studies will be needed not only to refine risks of premenopausal breast cancer associated with these mutations in cancer predisposition genes, but also to allow researchers to identify well-established mutational signatures characteristic of this special population. Lastly, since we employed gene panel testing, that allows for screening of multiple potentially clinically relevant genes, large genomic rearrangements in the familial breast and ovarian cancer genes were not reported in our work. However, this analysis contributes to a limited number of studies that provide information on the frequency of germline and somatic mutations in cancer predisposition genes found in premenopausal breast cancer, utilizing the powerful NGS technology.

In conclusion, etiology of premenopausal breast cancer was associated with germline mutations in 42.5% of cases, with 71.4% of tested patients from our cohort carrying somatic mutations in known cancer predisposition genes. The data presented in our work, and similar data obtained from expanded germline panel testing in clinical settings, provide the framework towards enhancing our understanding of premenopausal breast cancer etiopathogenesis and establishing connections between germline alterations and cancer risk in specific populations. Given the high incidence of gene mutations, genetic testing could eventually benefit not only the treatment of premenopausal breast cancer patients, but also future prevention and control strategies of secondary malignancies in the affected individual, as well as guidance of at-risk family members.

## Figures and Tables

**Table 1 genes-13-01362-t001:** Histopathological Status of Premenopausal Breast Cancer Tumors.

	Parameters	Numbers (%)
Age range	20–29	3 (5.6%)
	30–39	20 (37.0%)
	40–49	27 (50.0%)
	50–52	4 (7.4%)
Histological type	IDC	52 (96.3%)
	ILC	2 (3.7%)
	Other (medullary, inflammatory, metaplastic)	0 (0%)
Tumor grade	Low (grade I)	2 (3.7%)
	Intermediate (grade II)	15 (27.8%)
	High (grade III)	37 (68.5%)
Tumor stage	IA	13 (24.1%)
	IIA	9 (16.7%)
	IIB	8 (14.8%)
	IIIA	11 (20.4%)
	IIIB	1 (1.9%)
	IIIC	7 (13%)
	IV	5 (9.3%)
Tumor size (T)	T1 (≤2 cm)	21 (38.9%)
	T2 (> 2cm but ≤5 cm)	18 (33.3%)
	T3 (>5 cm)	7 (13.0%)
	T4	3 (1.9%)
	Tx	4 (7.4%)
	N/A	1 (1.9%)
ER	Negative	12 (22.2%)
	Positive	42 (77.8%)
PR	Negative	14 (25.9%)
	Positive	40 (74.1%)
HER2	Negative	39 (72.2%)
	Positive	15 (27.8%)
Molecular Subtype	Luminal A	13 (24.1%)
	Luminal B (HER2 negative)	19 (35.2%)
	Luminal B (HER2 positive)	11 (20.4%)
	HER2 enriched	4 (7.4%)
	Triple negative	7 (13%)
Lymph Nodes	N0	21 (38.9%)
	N1	13 (24.1%)
	N2	8 (14.8%)
	N3	6 (11.1%)
	Nx	5 (9.3%)
	N/A	1 (1.9%)
Family History	Yes	21 (38.9%)
	No	33 (61.1%)

**Table 2 genes-13-01362-t002:** Germline and Somatic mutations identified in premenopausal breast cancer cases.

Gene with Germline Mutation	Nucleotide Change (cDNA)	Codon Change (Protein)	Clinical Significance	Somatic Mutation Pathogenic	Somatic Mutation VUS	Patient Number
*BRCA1* (Chr17)	c. 5266dup	p. Gln1756Profs*74	Pathogenic	(−)	(−)	13
	c. 3649T>C	p. Ser1217Pro	VUS	(−)	(−)	29
	c. 3700_3704del	p. Val1234Glnfs*8	Pathogenic	N/A	N/A	32
*BRCA2* (Chr13)	c. 352C>T	p. Arg118Cys	VUS	N/A	N/A	32
	c. 8386C>T	p. Pro2796Ser	VUS	N/A	N/A	34
	c. 1342C>T:	p. Arg448Cys	VUS	N/A	N/A	30
	c. 9613_ 9614delinsCT	p. Ala3205Leu	VUS	(−)	*BRCA2* c. 9613_ 9614delinsCT	31
	c. 9867T>G	p. Phe3289Leu	VUS	*TP53* c. 85_86del	*BRCA2* c. 9867T>G	33
					*ROS1* c. 433A>C	
*CHEK2* (Chr22)	c. 1232G>A	p. Trp411*	Pathogenic	N/A	N/A	14
	c. 470T>C	p. Ile157THr	Pathogenic	N/A	N/A	32
	c. 470T>C	p. Ile157THr	Pathogenic	(−)	(−)	39
	c. 480A>G	p. Ile160Met	VUS	(−)	(−)	36
	c. 1175C>T	p. Ala392Val	VUS	N/A	N/A	38
	c. 190G>A	p. Glu64Lys	VUS	(−)	*FGFR1* c. 2192_*del	37
*TP53* (Chr17)	c824G>A	p. Cys275Tyr	Pathogenic	*TP53* c. 824G>A	*ROS1* c. 433A>C	8
					*RET* c. 1684A>T	
	(−)	(−)	(−)	*TP53* c. 824G>A	KMT2C c. 7826G>A	17
					RB1 c. 1988A>G	
					NOTCH1 c. 2453T>C	
	(−)	(−)	(−)	*TP53* c. 424_433del	(−)	3
	N/A	N/A	N/A	*TP53* c. 488A>G	*RAD50* c. 1094G>A	44
					*TP53* c. 562_ 564delInsAA	
					*CCND1* c. 724-2A >C	
	N/A	N/A	N/A	*TP53* c. 586C>T	(−)	46
	(−)	(−)	(−)	*TP53* c. 743G>A	*KMT2C* c. 4873G>A	7
	(−)	(−)	(−)	*TP53 c. 743G>A*	*ROS1* c. 500G>A	27
				*PIK3CA c. 3140A>G*		
	(−)	(−)	(−)	*TP53* c. 990del	*MET* c. 4090C>T	9
					*NF1* c. 563C>A	
	N/A	N/A	N/A	*TP53* c. 818G>T	(−)	49
				*PIK3CA* c. 115G>A		
*RAD51C* (Chr17)	c. 80T>C	p. Leu27Pro	VUS	N/A	N/A	54
*RAD51D* (Chr17)	c. 412A>C:	p. Asn138His	VUS	(−)	*BRCA2* c. 9613_ 9614delinsCT	31
*BRIP1* (Chr17)	c. 2285G>A	p. Arg762His	VUS	N/A	N/A	35
*MUTYH* (Chr1)	c. 452A>G	p. Tyr151Cys	Pathogenic	(−)	(−)	13
*PMS2* (Chr7)	c. 1999G>A	p. Glu667Lys	VUS	(−)	(−)	53
*AKT1* (Chr14)	(−)	(−)	(−)	*AKT1 c. 49G>A*	*PTEN* c. 481A>G	16
					*RAD50* c. 443A>G	
ATM (Chr11)	c. 7475T>G	p. Leu2492Arg	VUS	N/A	N/A	28
	(−)	(−)	(−)	*AΤΜ* c. 494T > G	*ATM* c. 482A>C	15
*ROS1* (Chr6)	N/A	N/A	N/A	(−)	*ROS1* c. 433A>C	48
					*NOTCH1* c. 7487_7489del	
*PTEN* (Chr10)	(−)	(−)	(−)	*PTEN c. 99_100ins*	*FGFR3* c. 2272G>A	12
					*ROS1* c. 1135C>G	
					*BRAF* c. 1159G>A	
					*TP53* c. 691del	
	N/A	N/A	N/A	*PTEN c. 923delG*	*(−)*	55
				*PTEN c. 923delA*		
				*PIΚ3CA c. 1633G>A*		
PIK3CA (Chr3)	(−)	(−)	(−)	*PIK3CA* c. 3140A>G	*BRCA1* c. 3743C>T	5
				*PIK3CA c. 1090G>A*	*TP53* c. 622G>A	
	N/A	N/A	N/A	*PIK3CA* c. 1624G>A	*NF1* c. 7301T>C	41
	N/A	N/A	N/A	*PIK3CA* c. 3140A>G	*RB1* c. 43G>A	43
	N/A	N/A	N/A	*PIK3CA* c. 3140A>T	*TP53* c. 815T>C	45
					*IDH1* c. 949C>T	
					*CTNNB1* c. 1206del	
	N/A	N/A	N/A	*PIK3CA* c. 1624G>A	*RAD50* c. 2604T>G	50
	N/A	N/A	N/A	*PIK3CA c. 3140A > G*	*RAD50 c. 980G>A*	51
				*ERBB2 c. 2264T>C*	*TP53 c.304A>T*	
					*NF1 c. 2573C>G*	
	N/A	N/A	N/A	*PIΚ3CA c. 1633G > A*	(−)	55
				*PTEN c. 923delG*		
				*PTEN c. 923delA*		

Abbreviations: N/A: not available; Chr: Chromosome.

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
