# Peer review of "Multi-Gene Mutation Profiling by Targeted Next-Generation Sequencing in Premenopausal Breast Cancer"

_genes, 2022, doi:10.3390/genes13081362_

Round 1

Reviewer 1 Report

The article submitted by Dr. Zografos et al reveals the results of a Greek study on the genetic contribution to premenopausal breast cancer susceptibility. Indeed, the authors performed a NGS analysis on blood and/or tissue samples of 54 premenopausal breast cancer patients. 

The retrospective analysis has a descriptive intent. The article is well written and fluent. Please consider the following comments in order to improve the quality of the work:

-- Methods:

1) Please consider adding the word "retrospective" to cohort study if suitable for your protocol. (line 94)

2) How do authors select patients included in the analysis? Consecutive cases? Please consider adding this information in the methods section. (line 94). 

3) Please define the time of recruiting of the patients. (i.e. "In this retrospective cohort study, 54 incident consecutive cases of premenopausal breast cancer patients, recruited between ... and ..., underwent ....) (line 94)

4) Please consider adding the definition of ER+ and PgR+ tumours. 

5) Please consider adding a paragraph discussing the statistical methods. Which platform was used for collection of clinical data and analysis? (SPSS, Microsoft Excel, Microsoft Access, R..)

-- Results

1) Add T4 in brackets in line 151

2) Table 1: Please consider adding data about:
- how many tumours were luminal A, luminal B, TNBC, triple positive (ER+, PgR+, HER2+);
- pathological stage according TNM
- family history of cancer

Moreover, consider adding "(-)" in the keys.

3) Please consider adding a figure summarizing the results for each patient. Example: patient number 32 harbour a germline variant of BRCA1 and BRCA2, CHEK2. What was the biology of her tumour?

This is also in order to support and clarify what author state in the discussion about the "two hit" model. 

-- Discussion

1) Please consider adding a paragraph in the discussion section, in order to discuss studies about molecular profiling of premenopausal breast cancer. One example could be represented by https://doi.org/10.1371/journal.pone.0210372

Author Response

The article submitted by Dr. Zografos et al reveals the results of a Greek study on the genetic contribution to premenopausal breast cancer susceptibility. Indeed, the authors performed a NGS analysis on blood and/or tissue samples of 54 premenopausal breast cancer patients. The retrospective analysis has a descriptive intent. The article is well written and fluent.

We would like to thank the reviewer for his positive feedback and for the time devoted to improving our manuscript.

Please consider the following comments in order to improve the quality of the work:

-- Methods:

1) Please consider adding the word "retrospective" to cohort study if suitable for your protocol. (Line 94)

Thank you for your useful suggestion. We specified the retrospective design of our study in Line 97, as suggested.

2) How do authors select patients included in the analysis? Consecutive cases? Please consider adding this information in the methods section. (Line 94). Please define the time of recruiting of the patients. (i.e. "In this retrospective cohort study, 54 incident consecutive cases of premenopausal breast cancer patients, recruited between ... and ..., underwent ....) (line 94)

We thank the reviewer for pointing this out. Clarifications have been added in Lines 100-103.

4) Please consider adding the definition of ER+ and PgR+ tumours.

According to the guideline recommendations of the American Society of Clinical Oncology and College of American Pathologists Panel for Immunohistochemical Testing of Estrogen and Progesterone Receptors in Breast Cancer, as well as the latest edition of the American Joint Committee on Cancer’s Staging System for Breast Cancer, the cutoff to distinguish “positive” from “negative” cases should be ≥ 1% ER-positive tumor cells, which is in accordance with the criteria used for the assessment of the tumor samples of our clinic-based cohort. This information has been added in Lines 109-111, according to your valuable suggestion.

5) Please consider adding a paragraph discussing the statistical methods. Which platform was used for collection of clinical data and analysis? (SPSS, Microsoft Excel, Microsoft Access, R..)

Once again, we appreciate your constructive input. In the “Methods” section of the revised manuscript we added a paragraph mentioning the statistical methodological features of our descriptive analysis (Lines 154-157).

Results

1) Add T4 in brackets in line 151

2) Table 1: Please consider adding data about:

- how many tumours were luminal A, luminal B, TNBC, triple positive (ER+, PgR+, HER2+);

- pathological stage according TNM

- family history of cancer

Moreover, consider adding "(-)" in the keys.

Thank you for your observations. Table 1 has been modified accordingly, to include information concerning the intrinsic molecular subtypes at the time of initial breast cancer diagnosis, the TNM Stage, and the percentage of participants that presented with a family history of cancer.

3) Please consider adding a figure summarizing the results for each patient. Example: patient number 32 harbour a germline variant of BRCA1 and BRCA2, CHEK2. What was the biology of her tumour? This is also in order to support and clarify what author state in the discussion about the "two hit" model.

According to your helpful comment, Supplementary Table 1 has been added to present in greater detail the distinct clinicopathological characteristics of each premenopausal patient that participated in our study in relation to their NGS results.

-- Discussion

Please consider adding a paragraph in the discussion section, in order to discuss studies about molecular profiling of premenopausal breast cancer. One example could be represented by https://doi.org/10.1371/journal.pone.0210372

We revised the Discussion Section to include a more thorough commentary of the molecular profile of premenopausal breast cancer as presented in the literature in correlation with the most commonly identified in our cohort somatic pathogenic mutations (Lines 245-252 and 256-261).

Once again thank you for your consideration and for revising our manuscript. We believe that the reviewers’ comments are an important addition to our paper and thank you for helping us conceive it. We will be happy to address any potential future questions.

Reviewer 2 Report

the study highlights the use of NGS for profiling multiple genes in premenopausal breast cancer. Most of the comments are geared towards a better understanding of the data presented.

Introduction: Could the authors offer a brief explanation as to why these particular genes were chosen? (lines 66-68)

Results: Could the authors please define young-onset (what age range was considered as young-onset age?) (line 106)

It would be beneficial for the readers if the authors could include a brief description of how the germline and somatic mutations were identified respectively (i.e ref 40 and ref 34)

line 165: I am not sure if 3 of 17 could be considered as 'the most commonly mutated gene'. Would it be possible for the authors to rephrase this?

Line 173: Could the authors define pathogenic somatic mutations? are they the same as the somatic mutations defined in line 158?

Table 1: The majority of the patients were in the age range of 30-39 and 40-49. Have the authors observed any patterns of mutations in these specific age groups? If yes, could the authors include it in the results section?

The authors talk about predisposition to certain mutations. However, the data seems to show that there are different mutations in different patients. How do the authors think the panels will be designed for the testing if there are no strong patterns visible?

Author Response

the study highlights the use of NGS for profiling multiple genes in premenopausal breast cancer. Most of the comments are geared towards a better understanding of the data presented.

We thank the reviewer for thoroughly reading our work and for their time and effort in reviewing our manuscript.

Introduction: Could the authors offer a brief explanation as to why these particular genes were chosen? (Lines 66-68)

We thank the reviewer for pointing this out. The genetic testing of these genes is recommended by the latest NCCN Guidelines, based on strong evidence of an association with Genetic/Familial breast cancer risk and clarifications have been added to the main manuscript under the Introduction section (Lines 68-71).

Results: Could the authors please define young-onset (what age range was considered as young-onset age?) (line 106)

Young age at breast cancer diagnosis generally refers to women younger than 40, although no consensus has been reached, and various studies are using several age cutoffs. As you rightly noticed, the term young-onset does not accurately describe the population included in our study. Therefore, a correction has been made in the revised version of our manuscript to omit this term and replace it with the word “premenopausal” (Lines 111-112).

It would be beneficial for the readers if the authors could include a brief description of how the germline and somatic mutations were identified respectively (i.e ref 40 and ref 34)

In order to further highlight that next-generation sequencing (NGS) was performed in our study by a targeted gene-sequencing panel, clarifications have been added in Lines 259-260 and 295, according to your helpful suggestion.

 line 165: I am not sure if 3 of 17 could be considered as 'the most commonly mutated gene'. Would it be possible for the authors to rephrase this?

Thank you for your observation, we have revised the sentence in Lines 182 and 276-277.

 Line 173: Could the authors define pathogenic somatic mutations? are they the same as the somatic mutations defined in line 158?

According to the Guidelines of the American College of Medical Genetics and Genomics variants relevant to Mendelian disease are classified as (1) pathogenic, (2) likely pathogenic, (3) uncertain significance, (4) likely benign, or (5) benign. In our study, Line 158 refers to the 34 premenopausal breast tumor samples that were analyzed by NGS, in order to identify somatic mutations, whereas Line 173 refers to the 21 out of the 34 patients that eventually were found to harbor pathogenic somatic mutations, according to the above-mentioned ACMG classification. A definition for sequence pathogenic variants as proposed by the ACMG has been added in Lines 148-153, along with the supporting reference (ref.27).

Table 1: The majority of the patients were in the age range of 30-39 and 40-49. Have the authors observed any patterns of mutations in these specific age groups? If yes, could the authors include it in the results section?

We initially attempted to compare the mutational profile of younger premenopausal females with breast cancer (our cut-off point was set at below 40 years of age) compared to older premenopausal counterparts from single institution cohort. However, no statistically significant differences were identified between the two groups; yet the sample size is relatively small to extract a safe conclusion and thus we chose to present a descriptive analysis of our clinic-based series that we believe is a noteworthy addition to the current literature due to the challenges that this age-group faces and the application of the powerful NGS analysis.

The authors talk about predisposition to certain mutations. However, the data seems to show that there are different mutations in different patients. How do the authors think the panels will be designed for the testing if there are no strong patterns visible?

We thank the reviewer for this viewpoint, and we agree. The introduction of multi-gene panel testing has rapidly altered the clinical approach to hereditary breast cancer testing of at-risk patients. Based on next-generation sequencing (NGS) technology, these tests simultaneously analyze a set of known cancer predisposition genes, they are often more cost-effective, and increase the yield of detecting a pathogenic variant in a gene that will impact medical management. However, commercially available tests differ in the specific genes analyzed, and many of them provide information from testing “intermediate” penetrant genes that are not necessarily clinically actionable and do not change risk management. In our study, we used such an extensive predesigned panel for the identification of germline mutations, namely the TruSight Cancer sequencing panel, that was developed to sequence 94 genes associated with a predisposition towards cancer (e.g., breast, colorectal). As more genes are tested, there is an increased likelihood of finding variants without clear clinical significance or VUSs, and potentially this might have influenced the ability to distinguish clear patterns in our population, as you rightly pointed out. Furthermore, the limited number of patients included in this descriptive study, originating from a single institution, might be responsible for the lack of identifying mutational patterns, and this is mentioned in the limitations of our study in the revised manuscript (Lines 341-345). In this context, we believe that our results provide an initial assessment of the mutational landscape of premenopausal breast cancer and even though no clear patterns were observed possibly due to the above-mentioned limitations, our findings highlight the need for a more personalized approach of individuals at high germline risk, such as young premenopausal breast cancer patients; these observations are discussed in the fifth paragraph of the Discussion Section (Lines 315-333).

Once again thank you for your consideration and for revising our manuscript. We believe that the reviewers’ comments are an important addition to our paper and thank you for helping us conceive it. We will be happy to address any potential future questions.